# Crafting a Rigorous, Clinically Relevant Large Animal Model of Chronic Myocardial Ischemia: What Have We Learned in 20 Years?

**DOI:** 10.3390/mps7010017

**Published:** 2024-02-19

**Authors:** Christopher R. Stone, Dwight D. Harris, Mark Broadwin, Meghamsh Kanuparthy, Sharif A. Sabe, Cynthia Xu, Jun Feng, M. Ruhul Abid, Frank W. Sellke

**Affiliations:** Department of Cardiothoracic Surgery, The Warren Alpert School of Medicine at Brown University, Providence, RI 02903, USA; ddharris@bidmc.harvard.edu (D.D.H.); mbroadwin@lifespan.org (M.B.); meghamsh_kanuparthy@brown.edu (M.K.); ssabe@bidmc.harvard.edu (S.A.S.); cxu2@lifespan.org (C.X.); jfeng@lifespan.org (J.F.); ruhul_abid@brown.edu (M.R.A.); fsellke@lifepan.org (F.W.S.)

**Keywords:** cardiovascular disease, myocardial ischemia, metabolic syndrome, large-animal model, drug discovery, metabolomics

## Abstract

The past several decades have borne witness to several breakthroughs and paradigm shifts within the field of cardiovascular medicine, but one component that has remained constant throughout this time is the need for accurate animal models for the refinement and elaboration of the hypotheses and therapies crucial to our capacity to combat human disease. Numerous sophisticated and high-throughput molecular strategies have emerged, including rational drug design and the multi-omics approaches that allow extensive characterization of the host response to disease states and their prospective resolutions, but these technologies all require grounding within a faithful representation of their clinical context. Over this period, our lab has exhaustively tested, progressively refined, and extensively contributed to cardiovascular discovery on the basis of one such faithful representation. It is the purpose of this paper to review our porcine model of chronic myocardial ischemia using ameroid constriction and the subsequent myriad of physiological and molecular–biological insights it has allowed our lab to attain and describe. We hope that, by depicting our methods and the insight they have yielded clearly and completely—drawing for this purpose on comprehensive videographic illustration—other research teams will be empowered to carry our work forward, drawing on our experience to refine their own investigations into the pathogenesis and eradication of cardiovascular disease.

## 1. Introduction

Of all pathophysiologic processes, there is none as deadly and none as ubiquitous as heart disease. This is true in the United States, where the most recent statistics collected by the American Heart Association and the Centers for Disease Control indicate that over 20 million Americans are afflicted with coronary heart disease (CHD); that a myocardial infarction occurs every 40 s; and that the age-adjusted death rate of heart disease is both higher than any other case and increasing in recent years [1,2]. Internationally, the data describe a similarly preeminent role for cardiovascular disease, with over 9 million deaths attributed to ischemic heart disease (IHD) recorded in 2021 alone and implicating this as the leading cause of death worldwide by a margin of nearly 2.5 [3]. This has translated in the United States to the performance of approximately 482,000 percutaneous and 202,000 surgical corrective procedures for coronary disease annually. Despite this substantial procedural volume, there are currently a large proportion of patients afflicted with CHD who lack revascularization options; this population has been estimated to reach as high as 1.8 million patients in North America alone [4]. The unparalleled breadth and the severity of the physiologic impact of heart disease constitute mandates to continue in the pursuit of improved and novel therapies [5,6,7,8,9,10,11,12,13,14]. The scientific community has responded robustly to this call, with the National Institutes of Health spending over USD 1 billion annually for the last 14 years on heart-disease-related research [15]. Although this is a small fraction of the total economic burden of heart disease, which has been estimated as lost income annually in the United States alone of USD 203.3 billion [16], it has nevertheless yielded numerous profoundly impactful advances in the prevention, diagnosis, and management of cardiovascular pathology [17].

As this necessary work continues, it is essential that it be conducted in a manner optimally suited to the translatability of significant basic scientific discoveries to the clinical setting; unfortunately, there have been many potential therapies that have failed to realize their preclinical promise in clinical trials [18]. Although the reason for this failure of translation is certainly multifactorial, one prominent cause is the failure of preclinical models to fully recapitulate the nuances of human cardiac anatomy, pathophysiology, and comorbidities. Such models, most commonly murine (e.g., ligation of the left anterior descending coronary artery to produce myocardial infarction), are an indispensable part of cardiovascular research given the technical, financial, and ethical challenges inherent in the use of larger organisms and have undoubtedly produced many significant insights into the fundamental pathobiology of heart disease [19]; still, the translational success that is urgently needed for the millions of patients suffering with these disease processes demands the discovery and utilization of the most faithful models possible.

In the case of IHD, which represents the major contributor to the overall burden of cardiovascular disease, swine are considered to represent the most suitable model organism, with suitability construed as potential for extrapolation to patients and in conformity to a number of criteria defined in the literature:Faithful representation of the disease process as it occurs in humans;Provision of experimental latitude over the location and chronicity of the induced ischemic insult;The induced insult is physiologically measurable and the results are reproducible;The model admits of the development of comorbidities usually associated with IHD in humans, such as metabolic syndrome;Amenability of the model to post-ischemic interventions, e.g., systemic or intracardiac administration of drugs, nanoparticles, and extracellular vesicles;The model is logistically feasible to administer in terms of technical difficulty, time required by the protocol, and cost associated with these considerations [20].

Pig models satisfy these criteria chiefly because of their close anatomic, physiologic, metabolic, and proteomic analogy with human cardiovascular disease. Specifically, the pig heart relative to its total body weight is similar to the proportions seen in humans; ventricular electromechanical function is similar between swine and humans; and, importantly, swine possess similar coronary arterial anatomy and pathophysiology to humans [21,22] but do not have a substantial collateral circulation, permitting the reliable, reproducible induction of ischemia through use of experimental means to occlude porcine coronary arteries. Moreover, swine share with humans the tendency to develop atherosclerotic plaques with age and to develop atherosclerotic lesions in response to dietary modification and hyperglycemia [23]. The logistical advantages provided by swine are also worthy of consideration, as they reach maturity more rapidly than other animal models and breed year-round, facilitating regular availability for scientific purposes [24].

It should also be noted that several additional species have been utilized throughout the years to study IHD, with varying degrees of conformity to the principles articulated above. One such species is the guinea pig, which has been utilized in the study of cardiovascular disease due to similarities in electrophysiological characteristics and cholesterol metabolism with humans [25]. Although this has permitted electrophysiologic insights to be derived from such models [26,27], the robust endogenous collateral coronary network possessed by guinea pigs differs from humans in providing the species with a corresponding bulwark against ischemic disease and thus limits their applicability to the study of chronic myocardial ischemia [28]. Another species utilized in small animal models is the rabbit, the hearts of which possess coronary anatomical and electrophysiological similarity to the hearts of humans, including a paucity of collateral circulation and a corresponding reliability of infarct induction [29]. Given this, as well as the analogous development of atherosclerotic lesions in rabbits to the process as it occurs in humans, it is not surprising that these animals have been used for many decades in cardiovascular research [30,31,32]. These considerations, in conjunction with the increased phylogenetic proximity of rabbits to humans, may warrant consideration of use of this organism as a bridge between small- and large-animal models of IHD [33]. Historically, the large animal of choice was the canine [34,35], and hemodynamic and size similarities to humans support this option, while the rich collateralization of the dog heart in conjunction with ethical objections argue against it [36].

Taken together, these considerations have proven compelling for many research teams invested in translational efforts to ameliorate the impact of CHD and have culminated in the consideration of swine as the most attractive model organism for the study of cardiovascular disease [36]. In the case of our research group, a porcine model of chronic myocardial ischemia has been in use, with elaboration aimed at maximizing experimental efficacy and facility of administration, for nearly three decades. The model over this time has proven to be extraordinarily fruitful, providing insights into a vast network of molecular underpinnings of the myocardial response to ischemic disease that have, in turn, served as the basis for numerous substantial advances in the pharmacologic armamentarium against ischemic insults. These achievements are summarized in tabular form below (Table 1), and references are provided for further elaboration of the state of the experimental model at the time it was used to produce them. It exceeds the scope of this protocol paper to fully delineate the alterations our model has undergone over the course of its development, but readers are invited to review the references included in Table 1 for additional explanation of features that do not appear in the contemporary configuration described below.

## 2. Experimental Design

As is readily apparent through the many modulations we have made to our protocol over the past several decades, the endpoints relevant to the study of ischemic cardiac disease may be successfully attained through a multitude of approaches; consider, for instance, the use of ultrasound, cardiac MRI, or pressure–volume loops for the assessment of myocardial functional responses to treatment. In this example, as well as across the other modifiable steps in our protocol, there are virtues and drawbacks to each possibility. With this section, we will present the essential components of our protocol together with rationales for their selection over potential alternatives, with the aim to aid researchers in understanding why the protocol delineated in the subsequent section exists in its current form [52,53,54,55,56]. It is our hope that this discussion will allow other teams to adapt our methods to their unique scientific goals and available resources.

### 2.1. Establishment of Chronic Myocardial Ischemia

Although our lab has employed a similar model to study acute myocardial infarction in the past, our current focus entails the induction of chronic myocardial ischemia using the surgical placement of an ameroid constrictor device (Research Instruments SW, Escondito, CA, USA). This technique, which has been employed by many investigative teams for over 60 years [57,58,59], generates a condition in the affected myocardial territory known as hibernating myocardium, in which blood flow and function are depressed, but the tissue remains viable and possesses reserve that may manifest with treatment aimed at amelioration of ischemia [60]. In addition to the clinical relevance of simulating the pathophysiology of the human coronary circulation when subjected to gradual atherosclerotic occlusion, this approach also entails the tolerability advantage of sparing a myocardium relatively deficient in epicardial collaterals from the sudden and total cessation of blood flow produced by ligation [61]. Ameroid constrictors consist of two major components: an outer, nondeformable rim of plastic or metal such as stainless steel or titanium and an inner ring composed of hygroscopic casein. When placed around the LCx, this inner casein ring undergoes gradual centripetal swelling, yielding correspondingly gradual coronary occlusion and ischemia to the affected myocardial territory [62]. This process produces reliable, rapid reductions in flow through the device over the first few postoperative days, with progression to complete occlusion occurring over the following four weeks, over which the collateral circulation develops. Placement of the constrictor over the LCx, which is the smallest of the three major coronary arteries in swine, confers the additional advantage of minimizing the extent of ventricular myocardium affected by ischemia to approximately 20%, which may mitigate the extent of sudden cardiac deaths due to dysrhythmia or infarction produced by this model [63,64]. This, in turn, permits survival of experimental animals for a duration sufficient to demonstrate differential degrees of pathophysiologic change produced by treatment. The duration of ischemia utilized in our protocol of 4 weeks is consistent with results obtained in human studies, in which early revascularization (<35 days) produced improved function in the ventricular myocardium, while waiting longer than this failed to do so [65]. Placement should be proximal to the first obtuse marginal branch of the LCx in order to produce a consistent ischemic region of sufficient size for subsequent tissue studies, and ameroid size should be tailored to that of the artery as visualized intraoperatively [61].

There are multiple alternatives to ameroid constriction utilized in the literature, including the production of fixed stenosis using silk ties or the serial placement of flow probes and hydraulic occluders [66,67,68]. Although these methods provide the advantages of precise calibration of the degree of stenosis and a reduction in myocardial blood flow at rest that may be desirable for studies focused on flow modulation, they are hampered both by occasionally high mortality and by increased technical demand [64]. In the case of hydraulic coronary occlusion, dissection of the vessel for 1–2 cm to accommodate serial placement of a flow probe and the occluder is needed, whereupon a cuff placed over the vessel and gradually inflated to the desired degree of stenosis. Similarly, silk tie occlusion involves dissection of an appropriate length of vessel free, whereupon a needle is placed over and parallel to the artery. Silk suture is then used to occlude the artery to the diameter of the needle after the needle is removed. A multitude of needles may be needed for this in order to progressively occlude the artery to achieve flow values consistent with a pre-specified target [66]. Advantages seen with such methods include inherent standardization of rate and degree of occlusion, including the option to modulate stenosis postoperatively in the case of hydraulic occluders if the device is externalized, while disadvantages include technical demand and comparative unfamiliarity in the field [64]. In contrast to the sophisticated equipment needed for and technical complexities of studies such as these, induction of chronic myocardial ischemia by means of an ameroid constrictor as described below is technically straightforward; consistently effective as assessed by measurement of differential myocardial blood flow; productive of a low mortality rate of approximately 20%; and, given its status as the most commonly employed model, optimal for comparison with historical results and across investigative teams [69].

### 2.2. Analysis of Myocardial Blood Flow Using Isotopic Microspheres

Following the placement of the ameroid constrictor device, confirmatory assessment of the affects thereof on coronary blood flow is necessary to confirm induction of ischemia and thus to validate any changes in the ischemic myocardium produced by treatment provided to animals in the experimental groups. To accomplish this, our group uses isotope-labeled microspheres (BioPhysics Assay Laboratory, Inc., [BioPAL], Worcester, MA, USA). The associated procedure involves the injection of any of ten available solutions of stable isotope microspheres, measuring approximately 15 μm and supplied at a concentration of 2.5 million microspheres per mL. Our lab routinely uses gold-labeled microspheres to map the ischemic territory, lutetium microspheres to determine blood flow at rest, and samarium microspheres for blood flow during rapid pacing; we also maintain a supply of several other isotopes, including europium, lanthanum, and ytterbium, should repetition of the assay become necessary. Briefly, gold microspheres are injected into the left atrial appendage during mechanical occlusion of the LCx with a vessel loop immediately prior to ameroid placement. Because the vessel is occluded, subsequent quantification of microsphere content in the myocardial territory supplied by the LCx will reveal a relative paucity of gold microspheres when compared with territories supplied by the other epicardial coronary arteries. Similarly, when lutetium and samarium are injected into the left atrial appendage during the subsequent harvest surgery, myocardial blood flow can be calculated, provided a constant rate of pump-mediated blood withdrawal (6.67 mL/min using the Harvard Apparatus, Holliston, MA, USA) according to the following equation: myocardial blood flow = (reference blood flow [mL/min]/tissue weight [g]) × (tissue microsphere count/reference blood microsphere count). In other words, the ratio of microsphere concentration in the tissue to the withdrawn blood is used to derive the rate of blood flow to the myocardial tissue sections sent for analysis. Because different isotopes are injected at rest and during rapid pacing and analyzed separately, any change in blood flow produced by these conditions is discovered using this technique. Assessment of blood flow at rest using lutetium reveals any change in myocardial blood flow that may arise due to the administration of treatments. The additional step of using samarium to track changes during rapid pacing is necessary because the collateral circulation that develops over the course of treatment restores blood flow to the occluded region during the weeks over which treatment is administered but is insufficient in the context of exercise-induced stress [70]. This physiologic nuance is clinically significant as it reflects the onset of anginal pain with exercise in patients [71]. Synthesizing these procedures, a total of three isotopes are injected into all experimental animals across the two surgical procedures we perform: gold during the ameroid constrictor surgery and lutetium and samarium during the terminal myocardial harvest surgery. At the end of the latter, the heart is resected and the left ventricle is sectioned into sixteen ways by dividing it first into apical and basal sections at the midpapillary level and then into even circumferential sections. Of these sections, the basal free wall is most likely to have been most affected by ischemia, given that it houses the territory of the proximal LCx. For all animals, we verify this by sending samples from ten sections adjacent to the free wall for gold microsphere quantification. Because microspheres were injected during LCx occlusion, the most ischemic segment will be that in which there is the lowest count of gold microspheres. Relative perfusion to this section in particular, as determined by samarium and lutetium microspheres as above, represents a powerful means of quantifying the collateral augmentation produced by pro-angiogenic therapies in conjunction with immunohistochemical visualization of the vessels themselves [39].

In all cases, isotopic microsphere analysis is achieved using neutron activation, in which isotopes are rendered active following tissue collection and thus emit radiation that can be precisely quantified. Neutron activation entails exposure to a neutron field, precipitating brief gamma emission by microsphere atoms and subsequent spectroscopic analysis with quantification of disintegrations per minute. It does not require destruction of the tissue, raising the possibility of additional subsequent analysis if needed. This technique is highly sensitive, with the capacity to detect even a single microsphere embedded in analyzed tissue. Additionally, it confers the substantial advantage over the previous gold standard of radioactive microspheres of mitigating exposure of lab personnel to the consequent occupational hazards and over the optical technique previously used by our laboratory of obviating additional tissue processing [72,73,74]. Additional details of microsphere use pertaining to our protocol are provided in the subsequent Procedure section of this manuscript.

### 2.3. Quantifying Functional Changes in the Ischemic Myocardium

In an animal model designed to elucidate strategies for optimizing clinical outcomes in the context of advanced ischemic cardiac disease, there are no endpoints more import than those pertaining to cardiac function. There are a multitude of assessment modalities available for this purpose, including, in decreasing order according to invasiveness, pressure–volume loop recordings, intracoronary flow pressure and flow monitoring, cardiac MRI, and echocardiography [75]. Each of these techniques provides a unique perspective on the myocardial response to disease and therapeutics and may be combined within a single study should augmented analytic breadth be desired. Because of its ease of use and familiarity, echocardiography is often employed for myocardial functional assessment in swine models; geometric limitations imposed by the swine pectoral morphology may be overcome through use of a transesophageal probe [76]. Parameters possible to acquire using this technique include chamber dimensions and volumes as well as derivate assessments of ventricular function, such as ejection fraction and cardiac output. The ubiquity and facility with which echocardiography is performed provide the additional advantage of the iterability of this technique: serial echocardiograms can track changes in myocardial function over time after induction of ischemia and in response to treatment [69,77,78]. The related technique of sonomicrometry, which involves the direct implantation of piezoelectric crystals into the ventricular myocardium, allows for excellent resolution but lacks the portability of standard ultrasonographic analysis [79].

Another technique capable of providing remarkable resolution in myocardial functional analysis in swine is cardiac MRI, which provides the additional advantage of reducing operator dependence in the acquisition of data. MRI images can be acquired conveniently during the same anesthetic event as other surgical procedures and provide details regarding coronary patency, myocardial wall thickness and circumferential strain, blood flow, tissue viability, and ejection fraction. Like echocardiography, cardiac MRI can also be repeated in the interest of acquiring pre-surgical baseline data prior to the induction of ischemia, to which changes attributable to treatment can then be compared [80,81].

Although echocardiography and cardiac MRI clearly possess strong advantages and have been employed by our lab in the past (see Table 1), we no longer use these techniques and have arrived instead in recent years at pressure–volume loop catheter-based transduction of myocardial functional parameters as our preferred analytic method. This is because, while the desirability of longitudinal functional analysis should not be understated, the concurrent acquisition of pressure- and volume-based parameters permits a load-independent assessment of cardiac chamber mechanics that cannot be obtained by other means [82]. Direct left ventricular placement of a pressure–volume loop catheter, as practiced in our lab, currently allows, in conjunction with aortic pressure acquisition by means of a separate catheter placed by femoral arterial cutdown, a comprehensive panel of functional myocardial parameters, including stroke volume, stroke work, ejection fraction, cardiac output, the pressure–volume area and myocardial oxygen consumption, the end-systolic pressure–volume relationship (ESPVR) and contractility, and the end-diastolic pressure–volume relationship (EDPVR) and ventricular compliance. To obtain the ESPVR and EDPVR, modulation of ventricular preload is necessary for the generation of multiple values; this is uniquely possible with invasive pressure–volume loop acquisition and easily achieved in our model through use of a vessel loop to occlude the inferior vena cava (IVC) [83]. Notably, it is also possible to perform this procedure using different cannulation configurations, including using a closed-chest approach involving balloon occlusion of the IVC, ventricular catheterization by way of the femoral artery for pressure transduction, and acquisition of volume measurements with cardiac MRI [84]. Given that our protocol entails myocardial harvest for tissue analysis immediately following pressure–volume loop generation, however, our lab has not integrated these less invasive alternatives into our procedure.

## 3. Procedure

In this section, we describe in detail the procedural steps comprising the induction and subsequent analysis of our porcine model of chronic myocardial ischemia. The first subsection is devoted to induction through surgical placement of an ameroid constrictor, as discussed above, while the second subsection outlines the second analytic surgical procedure, in which pressure–volume loop acquisition and myocardial harvest are performed. Although the time between these procedures and the age of the swine on which they are performed vary according to the analytic objective of a specific protocol, we typically perform these procedures on juvenile swine (at between 5 and 12 weeks of age), with approximately 6 weeks elapsing between procedures. The use of juvenile animals is necessary to prevent immediate coronary occlusion following ameroid placement, while the interval between procedures allows both for ameroid closure and the effects of any treatment administered to ramify on cardiac functional and tissue parameters subsequently studied. In future experiments, we hope to utilize aged animals, as this may more optimally model the impaired capacity for neovascularization that occurs in the elderly patients most commonly afflicted with ischemic cardiac disease [80,85]. A brief summary of the variety of data generated by the means of this protocol is presented in the subsequent section; specific examples pertaining to protocols we have completed in the past are widely available in printed and digital media, as exemplified in the references provided in Table 1.

### 3.1. Ameroid Constrictor Placement (1–2 h)

#### 3.1.1. Preoperative Phase

Swine receive aspirin (10 mg/kg) 1 day preoperatively and continue on it 5 days postoperatively. This increases consistency in ischemic region development [86].Anesthesia is induced with intramuscular injections of telazol (4.4 mg/kg) and xylazine (2.2 mg/kg). Buprenorphine (0.03 mg/kg) and a fentanyl patch (4 ug/kg) are given prior to surgery for analgesia.Following an aseptic prep, a 20–22-gauge catheter is inserted percutaneously into a large auricular vein. This is connected to a drip with 0.9% saline at a rate of 5–10 mL/kg/h. Excede (ceftiofur, a broad-spectrum cephalosporin antibiotic) is administered as a single dose IM preoperatively for antimicrobial prophylaxis.Intubation is performed with a cuffed endotracheal tube to establish an airway for mechanical ventilation. Isoflurane 0.75–3.0% is used for maintenance anesthesia. The pig is prepped and draped in the usual sterile fashion using an iodoform antibiotic solution and sterile cloth drapes.

#### 3.1.2. Intraoperative Phase

With the animal positioned supine, lidocaine 2% (2–5 mg/kg) is administered intradermally prior to the skin incision.A mini left thoracotomy is then performed: a transverse incision is made along the ribs beginning at the left sternal border and extending laterally for approximately 6 cm. This incision is placed just inferiorly to the left axillary fold (Figure 1).Electrocautery is used to cut through the skin and muscle of the chest wall. A Weitlaner retractor is used to retract the muscle. The intercostal muscles are divided just above the 3rd or 4th rib (Figure 2), and a Kelly clamp facilitates entry into the parietal pleura and protects the underlying intrathoracic structures as the incision is enlarged; finally, a rib spreader is placed to open the incision.The pericardium is opened (Figure 3) with scissors. It is important to identify and protect the phrenic nerve.The lingula of the left lung is identified and pushed back into the chest cavity if needed to avoid obstruction of the visual field.A 3-0 silk suture secures the pericardium to the skin, elevating the heart into the surgical field and retracting the pericardium (Figure 4).Twenty mL of blood is withdrawn from the left atrial appendage to be sent for baseline metabolic parameters prior to treatment, as needed. A Satinsky clamp is used to isolate the puncture (Figure 5), a 2-0 silk tie is placed around the clamp to ligate the puncture, and a small hemostat is applied to the tie pull the appendage out of the surgical view.Using Metzenbaum scissors, a small nick is made in the epicardial fat overlying the left anterior descending/left circumflex artery junction. The left circumflex artery is identified (Figure 6) and dissected free. A silicone vessel loop is placed around the left circumflex artery (Figure 7).
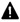
 CRITICAL STEP. Gentle retractile pressure is applied to the vessel loops to occlude the distal left circumflex coronary artery for 2 min; the EKG is monitored concurrently for ST segment elevations, confirming that the coronary circulation is successfully occluded. During this time, 5 mL of gold microspheres is injected by an assistant into the left atrial appendage. The injection apparatus is constructed as follows: a fine-gauge butterfly needle is affixed to a three-way stopcock, to which two syringes are also connected. One syringe contains the microsphere solution, while the other contains saline. The saline is used to confirm position within the atrial cavity by visualizing blood in the butterfly tubing upon withdrawal and also for flushing the tubing of microspheres after the microsphere syringe has been emptied. Microsphere injection transpires over 30 s and is begun shortly after occlusion of the artery. After 2 min, the pressure on the vessel loops is released. Confirmation of occlusion is critical as this is the means by which the ischemic territory is subsequently mapped, as described above.When the ST changes return to normal, the ameroid constrictor (1.5–2.5 cm, depending on size of the left circumflex artery, with 2.25 cm constituting the most commonly used device; see Figure 8) can be placed. Prior to attempting placement, it is essential to ensure that an adequate pocket is dissected: the proximal aspect of the vessel must be circumferentially freed from surrounding tissue, and the junction with the left main coronary artery should ideally be visualized. Small bridging veins may require clipping and division, and fine, non-absorbable suture (we use 6-0 polypropylene) should be made available in the event that vascular repair or ligation of venous bleeding is required. The device should be lubricated prior to placement to ease placement onto the artery. We have found that using Allis forceps to grasp the metallic outer casing of the ameroid permits a firm grip despite lubrication. An assistant may additionally use DeBakey forceps to retract surrounding epicardial tissue out from the intended ameroid pocket. Once an adequate pocket is visualized, the vessel loop is removed, and the ameroid is placed onto the artery. Forceps or a finger may be used to stabilize the ameroid while it is being placed, and the keyhole is rotated so that it faces outward (Figure 9). This will permit the device to remain in situ despite the beating of the heart (Figure 10), whereas the heartbeat may dislodge the device if it is facing in the opposite direction. If there is excessive manipulation of the left circumflex artery during this procedure, heparin will be given IV to help prevent thrombus formation. If there is spasm, topical nitroglycerin solution will be applied to the artery directly.

11.Prior to closure of the incision site, all aspects of the surgical field are carefully inspected for any bleeding and ligated as needed with suture or titanium hemostatic clips. If any injury to the lung is identified, it is repaired and appropriately tested to ensure no air leakage. These are uncommon but possible consequences of intrathoracic surgical procedures, and repairs are undertaken if any are indicated.12.The pericardium is closed with 3-0 Vicryl sutures. The ribs are closed with 0 nylon suture on a blunt-tipped needle. A breath-hold technique should be employed while placing the final rib suture, and a red rubber tube should be placed within the suture line and attached to wall suction during closure to evacuate air. Finally, the muscle and subcutaneous tissue are closed in layers with 3-0 Vicryl and 4-0 Monocryl suture, respectively.

#### 3.1.3. Postoperative Phase

Isoflurane is weaned, and veterinary staff assist with recovering the animal postoperatively in a padded recovery cage with heat support. The animal is returned to the housing room once stable and ambulatory.

### 3.2. Perfusion and Pressure–Volume Loop Acquisition, Terminal Myocardial Harvest (1.5–3 h, Figure 11)

#### 3.2.1. Preoperative Phase

Anesthesia is induced, and the animal is positioned, prepped, and draped in standard sterile fashion in an identical manner to that described above for the previous procedure. As this is not a survival surgery, neither aspirin nor antibiotic prophylaxis are necessary.

**Figure 11 mps-07-00017-f011:**
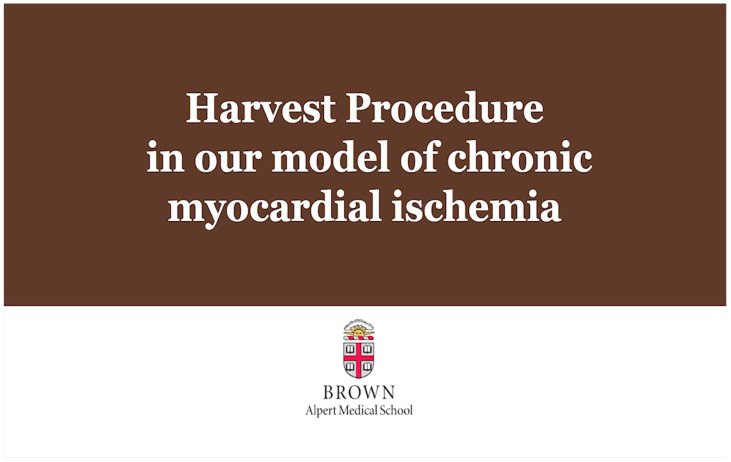
As this procedure has, to the knowledge of the authors, never been previously depicted, it is represented here in its entirety, rather than as isolated vignettes of critical steps as was performed for the ameroid constrictor procedure. Narration accompanies the video throughout to provide concise description of and rationale for all events while they are performed.

#### 3.2.2. Intraoperative Phase

Femoral arterial catheterization: after intubation and induction of general anesthesia, lidocaine is injected as above for the thoracotomy, but in this case, within the left or right inguinal crease, according to surgeon preference.A combination of sharp and blunt dissection using electrocautery and a right-angle clamp is utilized to expose the femoral artery, which will thus be cleared of surrounding tissue and encircled with vessel loops.Once the animal is heparinized (see below: the timing of this depends on progress made in the concurrently performed sternotomy by a co-surgeon), the Seldinger technique is used to obtain arterial access: an 18-gauge needle punctures the artery; a fine intravascular wire is placed through the needle; the needle is removed over the wire; and an appropriately sized dilator and sheath are advanced together over the wire to establish durable arterial access.Sternotomy, perfusion analysis, and pressure–volume loop acquisition: this procedure is initiated concurrently with femoral arterial catheterization by an additional surgeon. Electrocautery is used to incise the skin down to the bone between the sternal notch and xiphoid process.Blunt finger dissection is performed behind the sternum to permit space for a Lebsche knife, which is used to divide the sternum sharply down the midline. Use of a Lebsche knife is preferred over a sternal saw due to the sharp angulation and increased thickness of the superior portion of the porcine sternum; the knife permits readjustment as needed to divide through this region without risking damage to the underlying great vessels [87].The pericardium and pleura are dissected from the anterior chest wall with a combination of blunt and sharp dissection.A sternal retractor is placed to separate the rib cage and expose the heart.The pericardium is opened with Metzenbaum scissors.As pacing is needed to simulate a stressed heart and elucidate the physiologic consequences of collateral formation and the functional effects thereof, external pacemaker leads are attached to the left and right atria and tested to ensure capacity to pace the heart to 150 bpm as demonstrated by EKG.As IVC occlusion is needed to generate the ESPVR and EDPVR, a Satinsky clamp is used to encircle the intrathoracic IVC with a vessel loop for intermittent occlusion during pressure–volume loop acquisition.The animal is then heparinized, and femoral artery cannulation proceeds. The femoral arterial sheath is connected to the Harvard Apparatus withdrawal pump.Left atrial access is obtained using a fine-gauge butterfly needle attached to a three-way stopcock as performed for the ameroid constrictor surgery.Lutetium-labeled microspheres (5 mL) are then injected into the left atrium over 30 s while simultaneously withdrawing 10 mL of blood from the femoral artery over a period of 1.5 min at a rate of 6.67 mL/minute, as described above. This is performed at rest.Subsequently, samarium-labeled microspheres (5 mL) are injected into the left atrium and blood withdrawn from the femoral artery in the same fashion, but in this case during pacing at 150 bpm. Blood from Steps 13 and 14 is sent to BioPaL for perfusion analysis, along with left ventricular tissue samples (see below).Left ventricular apical access is obtained using Seldinger technique: a 3-0 polypropylene purse string suture is placed; an 18-gauge needle is positioned in the middle of the purse string; a wire is placed through the aperture and the needle removed; and a dilator and sheath are inserted and then tied in place using the polypropylene suture.A pressure–volume transduction catheter (Millar, Houston, TX, USA) will be inserted through this sheath, while another is inserted transfemorally. These catheters will be used to measure volumetric changes in the left ventricle and the pressure in the aorta. This procedure will be performed at resting heart rate, during IVC occlusion, and with pacing to 150 beats/min to yield a comprehensive survey of physiologic parameters for the assessment of treatment effects.After acquisition of pressure–volume data, the procedure is complete, and the surgeons proceed with myocardial resection and exsanguination under anesthesia, which is a method of euthanasia that is consistent with humane principles of animal care.All extraneous instruments, including electrocautery, suture, needles, hemostats, scissors, cannulation equipment, and pacemaker leads and clamps are removed from the field. The retractor remains in place to provide sufficient berth for myocardial resection.The superior vena cava (SVC) is dissected out from the surrounding tissue and the Kelly clamp. The IVC is already exposed and secured with a vessel loop from a previous step. A Satinsky clamp is placed to occlude the IVC, while a Kelly clamp occludes the SVC: this minimizes the effect of exsanguination on surgical site visibility.A #10 scalpel is used to remove all myocardial attachments, including the SVC, the IVC, the aorta, the pulmonary arteries, and the pulmonary veins.The resected heart is delivered from the thoracic cavity and brought to a separate table for sectioning. A series of sixteen circumferential sections defined in terms of proximity to the left anterior descending coronary artery are generated and snap-frozen in liquid nitrogen. Tissue from all other myocardial chambers, all cardiac valves, the great vessels, and a variety of peripheral organs are also resected and frozen for subsequent analysis.

## 4. Expected Results

See the references in Table 1 for a thorough overview of the discoveries underwritten by this model. In general, four categories of data are generated: 1. gold microsphere distribution results from samples of each of the 16 left ventricular sections taken to differentiate between ischemic and non-ischemic tissue for subsequent histologic, multi-omic, and other molecular analyses; 2. lutetium and samarium microsphere-mediated assessment of perfusion to all 16 ventricular sections, which can then be differentiated as ischemic or non-ischemic according to the aforementioned gold microsphere distribution results; 3. functional parameters derived from pressure–volume loop acquisition during the terminal myocardial harvest procedure as listed previously; and 4. a variety of molecular, histological, and multi-omic tissue analyses that serve the experimental objectives of the specific protocol [88,89,90,91,92]. As is clear from review of the results collected in Table 1, this model has proven extraordinarily fruitful over the past several decades of its operation in our lab; it is our sincere hope that this manuscript is employed to assist with the establishment of additional large-animal-based research programs designed to reveal the pathobiology of chronic myocardial ischemia. It is only with this foundation that cardiovascular research can continue its history of progress toward the generation of novel therapeutic strategies that will successfully ameliorate the impact of CHD on human morbidity and mortality.

## Figures and Tables

**Figure 1 mps-07-00017-f001:**
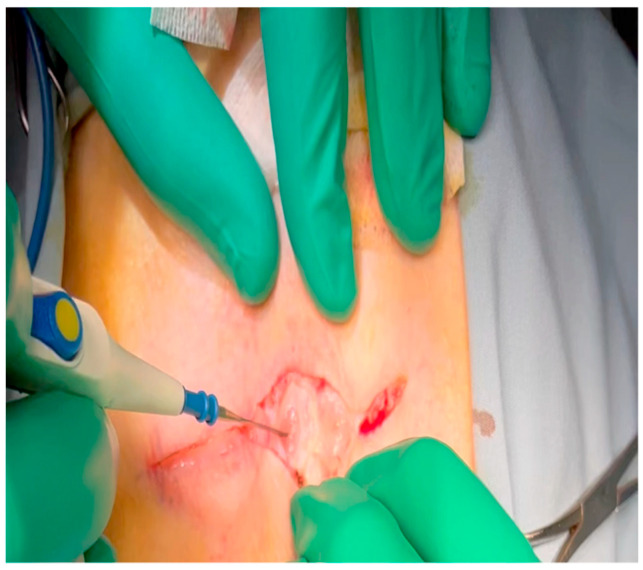
Electrocautery is used to make an incision through the subcutaneous tissue immediately inferior to the axillary crease.

**Figure 2 mps-07-00017-f002:**
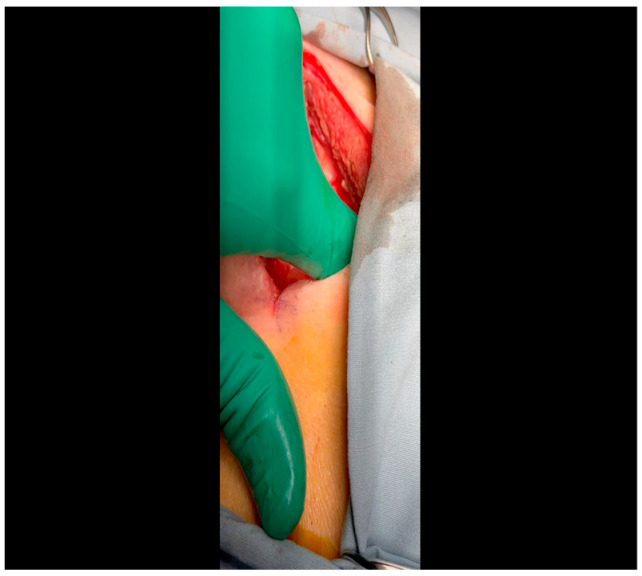
Rib spaces are counted to ensure entry into the thoracic cavity in a manner optimally suited for subsequent visualization of the LCx. This is achieved variably at the 3rd or 4th intercostal space.

**Figure 3 mps-07-00017-f003:**
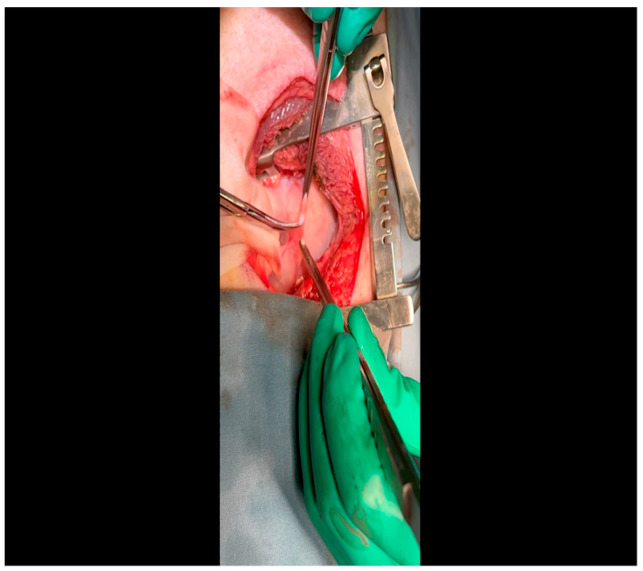
A pericardiotomy is made over the left atrial appendage; the assistant aids in this maneuver by tenting up the pericardial membrane above the underlying epicardial surface.

**Figure 4 mps-07-00017-f004:**
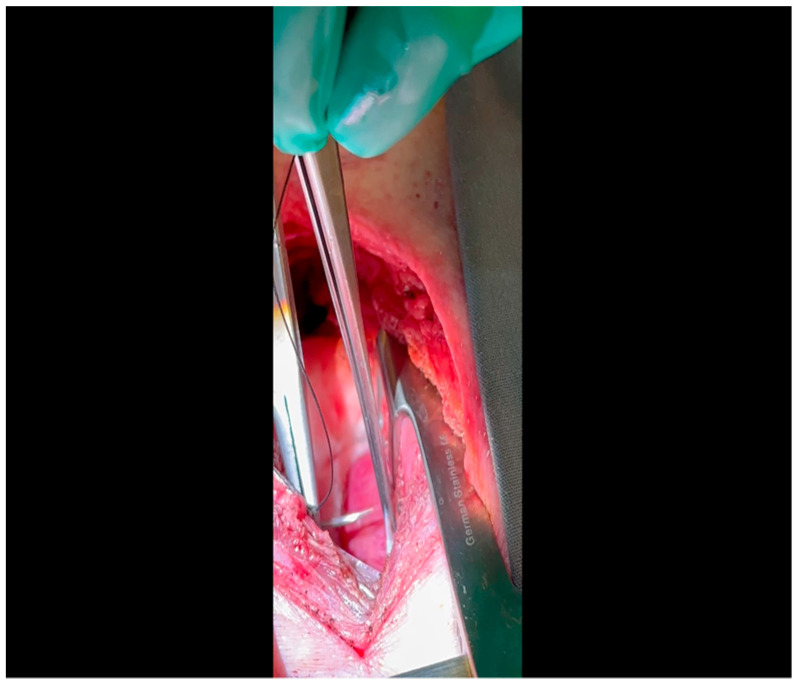
A fine silk stitch is driven first through the pericardium and then through the skin at the margins of the incision. This will elevate the heart into the field to approximate the LCx to the surgeon’s instruments.

**Figure 5 mps-07-00017-f005:**
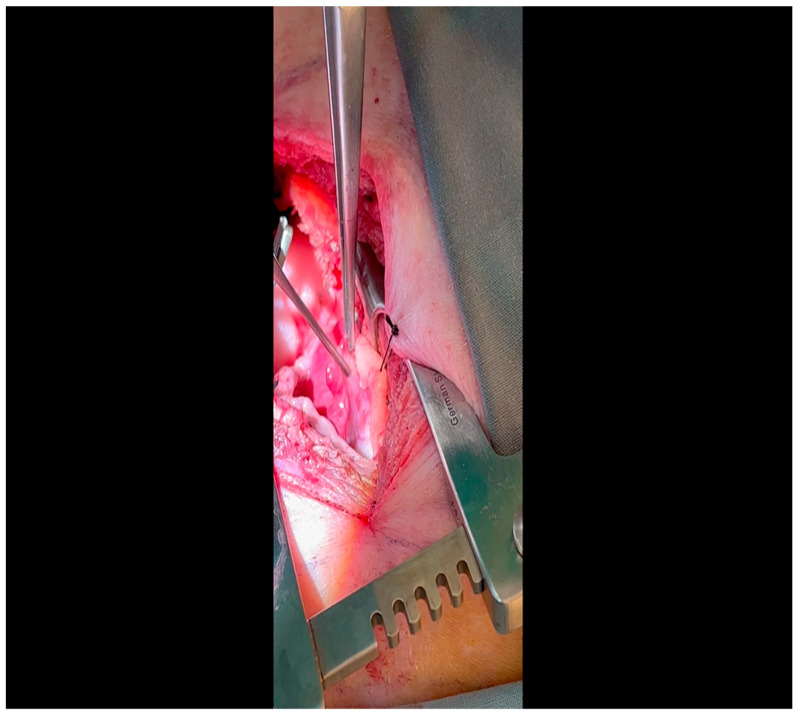
A Satinksy clamp is applied to the left atrial appendage at the site of puncture. A silk tie will be applied around this site, which will then serve the dual purposes of achieving hemostasis and aiding exposure.

**Figure 6 mps-07-00017-f006:**
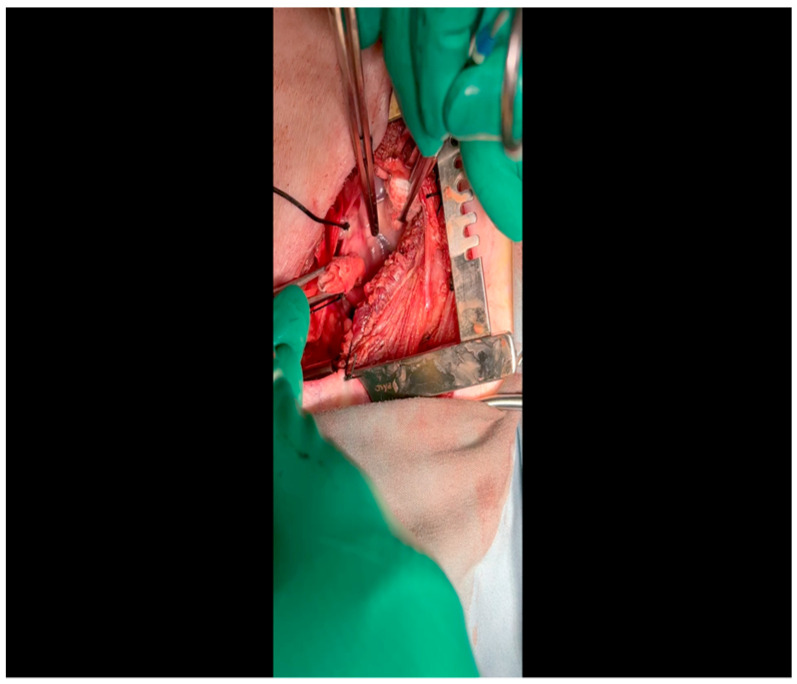
A silk tie retracts the left atrial appendage laterally; the assistant designates the presence of the LCx as it courses at the base of the atrioventricular groove.

**Figure 7 mps-07-00017-f007:**
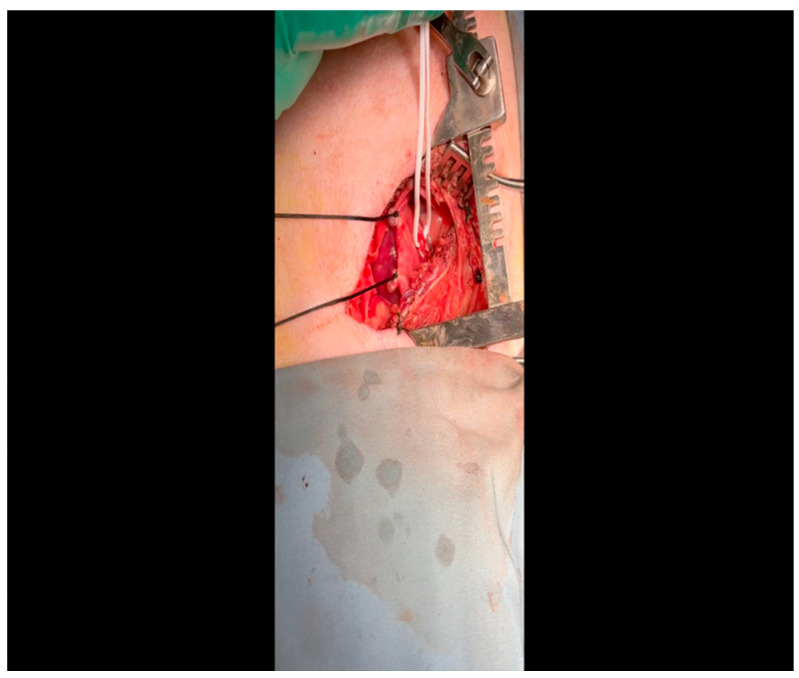
A vessel loop is placed around the LCx; an additional silk tie was applied to the left atrial appendage for optimal exposure.

**Figure 8 mps-07-00017-f008:**
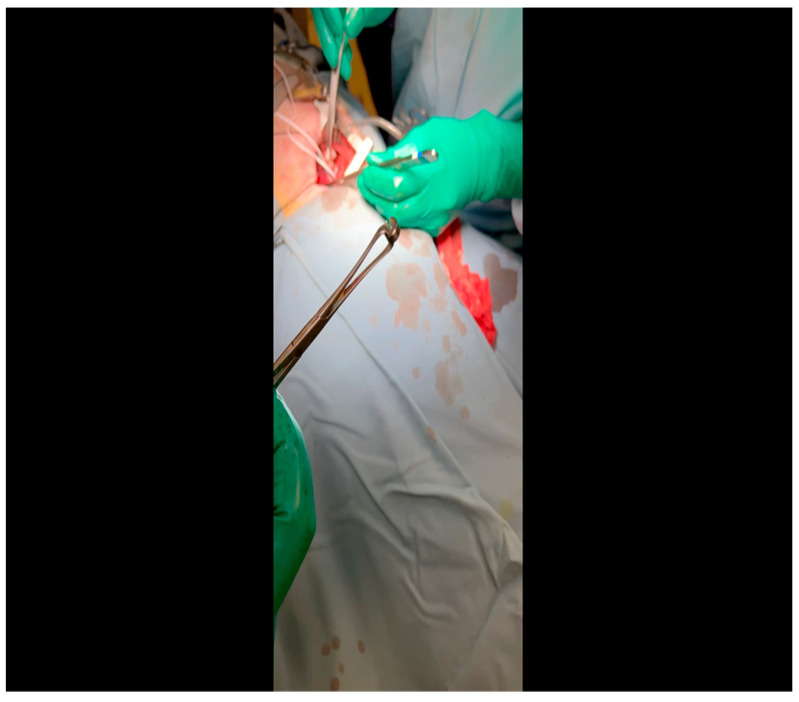
The ameroid constrictor is inspected to ensure that the outer metal and inner casein rings align. Lubricant may be applied to the ring to optimize the ease with which it is applied to the LCx.

**Figure 9 mps-07-00017-f009:**
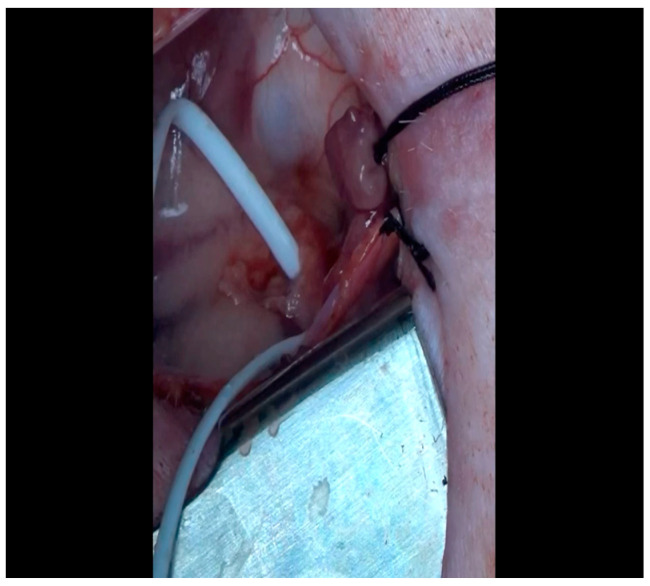
The vessel loop is removed and the lubricated ameroid constrictor is placed using Allis forceps around the vessel. The surgeon’s fingers and/or instruments are used to rotate the ameroid keyhole outward to prevent dislodgment due to the heartbeat.

**Figure 10 mps-07-00017-f010:**
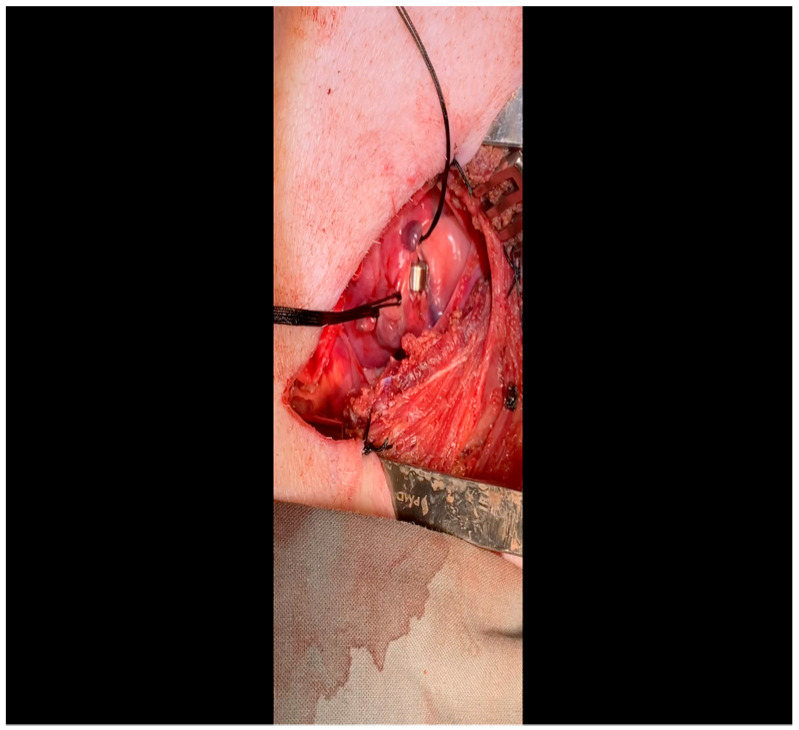
The ameroid constrictor is seen affixed to the LCx. Note that the keyhole is oriented outward as this minimizes the possibility of ameroid slippage prior to closure around the underlying vessel.

**Table 1 mps-07-00017-t001:** Although not comprehensive, this table provides a representative depiction of the insights achieved through use of our model; it also describes the variation of many key experimental components available to investigators aiming to design analogous protocol.

Project Title	Brief Description of Model	Results
Sodium-glucose Cotransporter-2 Inhibitor Canagliflozin Modulates Myocardial Metabolism and Inflammation in a Swine Model of Chronic Myocardial Ischemia (2023) [37]	Yorkshire swine underwent placement of an ameroid constrictor device over the proximal left circumflex coronary artery (LCx) to induce chronic ischemia in that vascular territory; experimental animals received oral canagliflozin treatment; and myocardial tissue was harvested and assayed for differential protein expression.	Canagliflozin treatment produced both inhibition of fatty acid oxidation and enhancement of insulin signaling in the ischemic myocardium, possibly providing accounts of the improved functional parameters seen with administration of this drug.
Extracellular Vesicle (EV) Treatment Partially Reverts Epigenetic Alterations in Chronically Ischemic Porcine Myocardium (2023) [38]	Yorkshire swine were fed either a regular or a high-fat diet before undergoing ameroid constriction of the LCx; two weeks later, all swine underwent redo thoracotomy for either intramyocardial injection of EVs or placebo. DNA was isolated and methylation profiling was performed.	DNA methylation was altered in the ischemic myocardium with dietary changes and by EV administration, providing mechanistic insight into the efficacy of EVs.
EVs Promote Arteriogenesis in the Chronically Ischemic Myocardium (2019) [39]	Yorkshire swine were fed a high-fat diet, after which they underwent ameroid constriction of the LCx. Two weeks later, they were given either intramyocardial vehicle, EVs alone, or EVs with calpain inhibition. Myocardial perfusion, function, and vascular density were assayed using isotopic microspheres, pressure–volume loop catheters, and immunohistochemistry, respectively.	Cardiac output, stroke volume, perfusion to the collateral-dependent myocardium, and arteriolar density were all improved after EV therapy.
Calpain Inhibition Decreases Myocardial Apoptosis in a Swine Model of Chronic Myocardial Ischemia (2015) [40]	Yorkshire swine fed a high-cholesterol diet underwent ameroid constrictor placement and received either no drug, low-dose calpain inhibition, or high-dose calpain inhibition; myocardial tissue was harvested and assayed for protein expressional changes.	Calpain upregulated angiogenic proteins and reduced apoptosis and oxidative stress in the ischemic myocardium.
Metformin Alters the Insulin Signaling Pathway in Ischemic Cardiac Tissue in a Swine Model of Metabolic Syndrome (2013) [41]	Ossabaw miniswine were fed regular or high-cholesterol diets and underwent ameroid constrictor placement over the LCx; treatment group animals were given metformin.	Metformin upregulated insulin signaling proteins in the ischemic myocardium, possibly accounting for the survival benefit of the drug.
Resveratrol Improves Myocardial Perfusion in a Swine Model of Hypercholesterolemia and Chronic Myocardial Ischemia (2010) [42]	Yorkshire swine were fed a regular or a high-cholesterol diet, an ameroid constrictor was placed around the LCx of all animals, and all subsequently underwent cardiac MRI and coronary angiography	With resveratrol, total cholesterol was significantly reduced in high-cholesterol diet animals, functional decrement was attenuated, perfusion was augmented, and angiogenic markers were elevated.
Insulin Enhances the Myocardial Angiogenic Response in Diabetes (2007) [43]	Yucatan miniature swine were given diabetes using alloxan and underwent ameroid constrictor placement around the LCx, with or without insulin therapy; isotopic microspheres were used to determine myocardial perfusion.	Diabetes precipitated endothelial dysfunction and impaired collateral-dependent perfusion associated with expression of angiogenic inhibitors; these deficits were all significantly mitigated with insulin.
Hypercholesterolemia Impairs the Myocardial Angiogenic Response in a Swine Model of Chronic Myocardial Ischemia (2006) [44]	Yucatan miniature swine were fed either a regular or a high-cholesterol diet and underwent ameroid-mediated LCx occlusion; isotopic microspheres were injected to determine myocardial perfusion and microvascular reactivity was studied in vitro.	Hypercholesterolemic swine exhibited endothelial dysfunction and reduced myocardial perfusion along with significantly increased endostatin expression (an inhibitor of angiogenesis).
Normalization of Coronary Microvascular Reactivity and Improvement in Myocardial Perfusion by Surgical Vascular Endothelial Growth Factor (VEGF) Therapy Combined With Oral Supplementation of L-arginine (2005) [45]	Yucatan miniature swine were fed either a regular diet, a high-cholesterol diet, or a high-cholesterol+L-arginine diet and subjected to ameroid constrictor-mediated LCx occlusion; then, three weeks later they were subjected to left thoracotomy for VEGF pump insertion and ultimately to isotope microsphere-mediated perfusion analysis, microvascular reactivity assays, and sonomicrometry for functional analysis.	High-cholesterol swine experienced endothelial dysfunction, which was reversed with L-arginine; L-arginine restored the effect of VEGF in high-cholesterol animals, improving endothelial cell density
Endogenous Myocardial Angiogenesis and Revascularization Using a Gastric Submucosal Patch (2003) [46]	Yorkshire swine underwent LCx occlusion, with or without subsequent gastroepiploic artery-based patch with transdiaphragmatic transfer and apposition to the ischemic territory.	Swine with patches exhibited higher perfusion with myocardial pacing and increased endothelial cell density in the ischemic territory.
A Novel Peroxynitrte Decomposer Catalyst (FP-15) Reduces Myocardial Infarct Size in an In Vivo Peroxynitrite Decomposer and Acute Ischemia-Reperfusion in Swine (2002) [47]	60 min ligation of the left anterior descending coronary artery followed by 180 min of reperfusion, with or without FP-15	Infarct size was reduced by 35% with FP-15 infusion.
Efficacy of Intracoronary or Intravenous VEGF_165_ in a Pig Model of Chronic Myocardial Ischemia (2001) [48]	Yorkshire swine underwent LCx occlusion which was then angiographically confirmed; they were then given either IV or intracoronary VEGF, the latter with or without nitric oxide synthase blockade; effects were assessed with repeat angiography and echocardiography.	Intracoronary nitric oxide synthase inhibition blocked VEGF-mediated hypotension; intracoronary VEGF improved myocardial collateralization
VEGF Administration in Chronic Myocardial Ischemia (1996) [49]	Yorkshire swine underwent LCx occlusion, followed by microcatheter-mediated delivery of VEGF; perfusion assessment with microsphere injection, functional analysis using pressure–volume catheter, direct embedding of ultrasonic crystals to assess regional wall motion	Improved blood flow to the ischemic territory with VEGF treatment, which resulted in better preservation of endothelium-dependent microvessel relaxation and left ventricular function
Cocaine and the Porcine Coronary Microcirculation: Effects of Chronic Cocaine Exposure and Hypercholesterolemia (1995) [50]	Yucatan miniature swine fed either a high-cholesterol or regular diet and administered intramuscular cocaine in the experimental group; cardiac tissue was extracted surgically and coronary microvessel were isolated and assayed for contractile response under a variety of conditions	Cocaine significantly reduces coronary microvascular diameter and blunts ß-adrenoreceptor-mediated relaxation
Basic Fibroblast Growth Factor (bFGF) Improves Myocardial Function in Chronically Ischemic Porcine Hearts (1994) [51]	LCx occlusion in Yorkshire swine, followed by bFGF administration in the experimental group; surgical acquisition of pressure–volume relationship data using intramyocardial catheter placement; perfusion assessed using dyed microspheres	Fourfold reduction in left ventricular infarct size, improved coronary blood flow, and improved resilience to hemodynamic insults

## Data Availability

No new data were created or analyzed in this study. Data sharing is not applicable to this article.

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
