# Peer review of "Crafting a Rigorous, Clinically Relevant Large Animal Model of Chronic Myocardial Ischemia: What Have We Learned in 20 Years?"

_mps, 2024, doi:10.3390/mps7010017_

Round 1

Reviewer 1 Report

Comments and Suggestions for Authors

A well written description of a complex model. I believe this will be of interest to readers as many scientific methods are described.

Minor comment: The list presented in lines 67-75 could be presented in list format to make it easier to read.

Comments on the Quality of English Language

Well written.  May consider using "swine" instead of "pigs" in the manuscript

Author Response

Thank you very much for the kind and timely feedback. We agree that 'swine' is a superior substitute for 'pigs' in the manuscript given the formality of the format; we have accordingly taken you up on this suggestion. Additionally, we have reformatted the lines you mentioned as a list separate from the main body of the text to maximize readability. Thank you again for time and help with improving the quality of our submission.

Reviewer 2 Report

Comments and Suggestions for Authors

Stone CR-mps-2828955

Comments:

The authors focused on, in their manuscript, accurate animal models for the refinement and elaboration of the hypotheses and therapies crucial to combat human disease. Indeed, the theme of this manuscript is related to the precise surgery techniques carried out in pigs, which model is the most accurate one in preclinical studies instead of small animals e.g., mice and/or rats. The authors’ experience to refine their preclinical research in the pathogenesis and eradication of cardiovascular diseases connected to myocardial ischemia, including diabetic subjects (e.g., metabolic syndromes). The research intensively started several decades ago (in 1980s) in dogs, monkeys and pigs, which cardiovascular anatomy (the heart) is close to humans, although to get nowadays a license on ‘dog or monkey research’ is difficult and/or almost impossible. Although, in 2023, a pig heart was transplanted in a human being, the final outcome was not as good as it was initially expected. However, this surgery was a good try.

The authors of this manuscript (CR-mps-2828955) concluded that cardiovascular investigation can continue its history of progress toward the generation of novel therapeutic strategies and interventions that may successfully reduce the impact of CHD on human morbidity and mortality.

-        The manuscript is very well written and methodically precisely designed. However, the checking of English (e.g., grammatical errors and typos) is not the duty of this reviewer. Some typos might be occurred throughout the manuscript, e.g., page 6, line 125: “… 60 years464748 generates a ……“. Checking of the manuscript is necessary.

-        The manuscript is evaluated on the scientific merit by this reviewer. The manuscript (CR-mps-2828955) is not a brand new, however, gives a very good and valuable picture about several technical questions step by step in connection with surgical methods in pigs.

Suggestions:

            How about the preclinical studies using rabbits and/or guinea pigs as “small animals”? It is generally accepted that these “small” hearts (guinea pigs and rabbits) are having several similarities to the human heart and have been used in preclinical studies for several years and decades. Therefore, this aforementioned theme (small animal hearts) should be discussed and some papers (see below) should be added and summarized as an additional paragraph in the revised version.

This reviewer believes that additional references (recent and ‘old’, as historical) are recommended to be involved and acknowledged (cited) in the revised version of the manuscript, since the incorporation of the list (see below: guinea pigs, rabbits, dogs) of published papers, in general, may substantially increase the interest of general readers, surgeons, senior and junior clinicians and experimental researchers.

Guinea pigs:

M. Siess, Some aspects on the regulation of carbohydrate and lipid metabolism in cardiac tissue Basic Res Cardiol, 1980 Jan-Feb;75(1):47-56. doi: 10.1007/BF02001393

Chaudhry FI, Dennis SC, Harness JB. J Biomed Eng. 1982 Oct;4(4):289-93. doi: 10.1016/0141-5425(82)90046-2

Brooks WM, Willis RJ. Determination of intracellular pH in the Langendorff-perfused guinea-pig heart by 31P nuclear magnetic resonance spectroscopy.

J Mol Cell Cardiol. 1985 Aug;17(8):747-52. doi: 10.1016/s0022-2828(85)80036-5

Moffat MP. Concentration-dependent effects of prostacyclin on the response of the isolated guinea pig heart to ischemia and reperfusion: possible involvement of the slow inward current. J Pharmacol Exp Ther. 1987 Jul;242(1):292-9 PMID: 2441027

M P MaxwellD J HearseD M Yellon, Species variation in the coronary collateral circulation during regional myocardial ischaemia: a critical determinant of the rate of evolution and extent of myocardial infarction, Cardiovasc Res, 1987 Oct;21(10):737-46, doi: 10.1093/cvr/21.10.737

Jianzhong AnAmadou K S CamaraSamhita S RhodesMatthias L RiessDavid F Stowe, Warm ischemic preconditioning improves mitochondrial redox balance during and after mild hypothermic ischemia in guinea pig isolated hearts, 2005 Jun;288(6):H2620-7.

 doi: 10.1152/ajpheart.01124.2004. Epub 2005 Jan 14.

Beaumont E, Southerland EM, Hardwick JC, Wright GL, Ryan S, Li Y, KenKnight BH, Armour JA, Ardell JL. Vagus nerve stimulation mitigates intrinsic cardiac neuronal and adverse myocyte remodeling postmyocardial infarction. Am J Physiol Heart Circ Physiol. 2015 Oct;309(7):H1198-206. doi: 10.1152/ajpheart.00393.2015. Epub 2015 Aug 14

Mendonca Costa C, Anderson GC, Meijborg VMF, O'Shea C, Shattock MJ, Kirchhof P, Coronel R, Niederer S, Pavlovic D, Dhanjal T, Winter J. The Amplitude-Normalized Area of a Bipolar Electrogram as a Measure of Local Conduction Delay in the Heart.

 Front Physiol. 2020 May 19;11:465. doi: 10.3389/fphys.2020.00465. eCollection 2020.PMID: 32508676

Rabbits:

Harken AH, Simson MB, Haselgrove J, Wetstein L, Harden WR 3rd, Barlow CH, Early ischemia after complete coronary ligation in the rabbit, dog, pig, and monkey. Am J Physiol. 1981 Aug;241(2):H202-10. doi: 10.1152/ajpheart.1981.241.2.H202

Wetstein L, Simson MB, Feldman PD, Harken AH. Pharmacologic modification of myocardial ischemia. Circulation. 1982 Sep;66(3):548-54. doi: 10.1161/01.cir.66.3.548.PMID: 7094266

Wetstein L, Rastegar H, Barlow CH, Harken AH. Delineation of myocardial ischemia in an isolated blood-perfused rabbit heart preparation. J Surg Res. 1984 Oct;37(4):285-9. doi: 10.1016/0022-4804(84)90190-2.PMID: 6482421

Vogel WM, Apstein CS. Effects of alloxan-induced diabetes on ischemia-reperfusion injury in rabbit hearts. Circ Res. 1988 May;62(5):975-82. doi: 10.1161/01.res.62.5.975

Mousa SA, Brown R, Thoolen MJ, Smith RD. Evaluation of the effect of azapropazone on neutrophil migration in regional myocardial ischaemia/reperfusion injury in rabbits. Br J Pharmacol. 1990 Jun;100(2):379-82. doi: 10.1111/j.1476-5381.1990.tb15813.x.PMID: 2165840

Simkhovich BZ, Kloner RA, Przyklenk K., Temporal changes in the subcellular distribution of protein kinase C in rabbit heart during global ischemia. Basic Res Cardiol. 1998 Apr;93(2):122-6. doi: 10.1007/s003950050072.PMID: 9601579

Kilgore KS, Park JL, Tanhehco EJ, Booth EA, Marks RM, Lucchesi BR. Attenuation of interleukin-8 expression in C6-deficient rabbits after myocardial ischemia/reperfusion. J Mol Cell Cardiol. 1998 Jan;30(1):75-85. doi: 10.1006/jmcc.1997.0573

Carter J, Buerke U, Rössner E, Russ M, Schubert S, Schmidt H, Ebelt H, Pruefer D, Schlitt A, Werdan K, Buerke M. Anti-inflammatory actions of aprotinin provide dose-dependent cardioprotection from reperfusion injury. Br J Pharmacol. 2008 Sep;155(1):93-102. doi: 10.1038/bjp.2008.223. Epub 2008 Jun 9.PMID: 18536753

Juhasz B, Kertész A, Balla J, Balla G, Szabo Z, Bombicz M, Priksz D, Gesztelyi R, Varga B, Haines DD, Tosaki A. (2013). Cardioprotective Effects of Sour Cherry Seed Extract (SCSE) on the Hypercholesterolemic Rabbit Heart. Curr Pharm Design, 2013;19(39):6896-905.

Chang SL, Hsiao YW, Tsai YN, Lin SF, Liu SH, Lin YJ, Lo LW, Chung FP, Chao TF, Hu YF, Tuan TC, Liao JN, Hsieh YC, Wu TJ, Higa S, Chen SA. Interleukin-17 enhances cardiac ventricular remodeling via activating MAPK pathway in ischemic heart failure. J Mol Cell Cardiol. 2018 Sep;122:69-79. doi: 10.1016/j.yjmcc.2018.08.005. Epub 2018 Aug 8

Awad K, Sayed A, Banach M. Coenzyme Q10 Reduces Infarct Size in Animal Models of Myocardial Ischemia-Reperfusion Injury: A Meta-Analysis and Summary of Underlying Mechanisms. Front Cardiovasc Med. 2022 Apr 15;9:857364. doi: 10.3389/fcvm.2022.857364. eCollection 2022

Baggett BC, Murphy KR, Sengun E, Mi E, Cao Y, Turan NN, Lu Y, Schofield L, Kim TY, Kabakov AY, Bronk P, Qu Z, Camelliti P, Dubielecka P, Terentyev D, Del Monte F, Choi BR, Sedivy J, Koren G. Elife. Myofibroblast senescence promotes arrhythmogenic remodeling in the aged infarcted rabbit heart. 2023 May 19;12:e84088. doi: 10.7554/eLife.84088

Dogs:

Dillmann WH, Mehta HB, Barrieux A, Guth BD, Neeley WE, Ross J Jr. Ischemia of the dog heart induces the appearance of a cardiac mRNA coding for a protein with migration characteristics similar to heat-shock/stress protein 71. Circ Res. 1986 Jul;59(1):110-4. doi: 10.1161/01.res.59.1.110.

Rossen RD, Michael LH, Kagiyama A, Savage HE, Hanson G, Reisberg MA, Moake JN, Kim SH, Self D, Weakley S. Mechanism of complement activation after coronary artery occlusion: evidence that myocardial ischemia in dogs causes release of constituents of myocardial subcellular origin that complex with human C1q in vivo. Circ Res. 1988 Mar;62(3):572-84. doi: 10.1161/01.res.62.3.572.PMID: 3257722

Moore PG, Reitan JA, Kien ND, White DA, Safwat AM. Role of systemic arterial pressure, heart rate, and derived variables in prediction of severity of myocardial ischemia during acute coronary occlusion in anesthetized dogs. Anesth Analg. 1992 Sep;75(3):336-44. doi: 10.1213/00000539-199209000-00004.PMID: 1510253

Dreyer WJ, Michael LH, Nguyen T, Smith CW, Anderson DC, Entman ML, Rossen RD. Kinetics of C5a release in cardiac lymph of dogs experiencing coronary artery ischemia-reperfusion injury. Circ Res. 1992 Dec;71(6):1518-24. doi: 10.1161/01.res.71.6.1518.PMID: 1423944

Ovize M, Przyklenk K, Hale SL, Kloner RA.
Preconditioning does not attenuate myocardial stunning.
Circulation. 1992 Jun;85(6):2247-54. doi: 10.1161/01.cir.85.6.2247.PMID: 1591839

Reffelmann T, Kloner RA. The "no-reflow" phenomenon: basic science and clinical correlates. Heart. 2002 Feb;87(2):162-8. doi: 10.1136/heart.87.2.162.

Kingma JG Jr, Simard D, Voisine P, Rouleau JR. Role of the autonomic nervous system in cardioprotection by remote preconditioning in isoflurane-anaesthetized dogs. Cardiovasc Res. 2011 Feb 1;89(2):384-91. doi: 10.1093/cvr/cvq306. Epub 2010 Sep 27.PMID: 20876586

Guo Y, Yang Q, Weng XG, Wang YJ, Hu XQ, Zheng XJ, Li YJ, Zhu XX. Shenlian Extract Against Myocardial Injury Induced by Ischemia Through the Regulation of NF-κB/IκB Signaling Axis. Front Pharmacol. 2020 Mar 6;11:134. doi: 10.3389/fphar.2020.00134. eCollection 2020.PMID: 32210797

Schneider SM, Sansom GT, Guo LJ, Furuya S, Weeks BR, Kornegay JN. Natural History of Histopathologic Changes in Cardiomyopathy of Golden Retriever Muscular Dystrophy. Front Vet Sci. 2022 Feb 17;8:759585. doi: 10.3389/fvets.2021.759585. eCollection 2021

Author Response

  1. Thank you very much for the multitude of perceptive suggestions to improve our submission. To address the first concern you raised, we have reviewed the manuscript again to ensure that its grammatical and typographical quality is maximized and have made all changes we identified in the interest of this.
  2. We appreciate your inclusion of these additional references and have added content concerning these models and references accordingly. Please see the attached file for a copy of the text we have drafted on the basis of your recommendations.

Reviewer 3 Report

Comments and Suggestions for Authors

1. Justification for Model Choice: While the authors justify using an ameroid constrictor device for inducing chronic myocardial ischemia, a more detailed comparison with other models and an explanation for choosing this method would strengthen the rationale.

2. In section 2 (Experimental Design), I suggest authors concentrate on the scientific literature related to flow measurement with microspheres and chronic ischemia, particularly ameroid constriction. The section seems more like a "discussioer than an experimental design, and there's confusion about mentioning MRI, which wasn't used in the protocol.

3. Ameroid Constrictor Placement Procedure: The section on ameroid constrictor placement is detailed, but there is room for further elaboration on critical steps. Especially in the video they provided, I would like to see truly encircling and placing the ameroid ring on the LCx. In addition, the authors could provide additional information on potential complications, troubleshooting strategies, and how to confirm the procedure's success.

4. Clarity in Isotopic Microsphere Analysis: The section on isotopic microsphere analysis is well-described, but some steps could benefit from more explicit details. For instance, the authors could elaborate on the preparation of isotopic microspheres, the specific methods used for injection, and the instrumentation and parameters used in neutron activation for analysis.

5. Notably, the authors indirectly measured flow and physiological changes, assuming ischemia without providing a measurement of the ischemic area. This aspect needs clarification. 

6. Surgical instrument: Does the lab where the authors work only have the Lebsche knife for sternotomy? Is there access to sternotomy saws? If so, what is the rationale behind using the knife?

Comments on the Quality of English Language

Minor editing of the English language required

Author Response

1. Thank you for your comment. While we intend for the focus of this work to be our protocol entailing ameroid constriction, we agree that additional discussion of the merits and drawbacks of the competing methods is warranted to assist readers in potential choices between methods. We have therefore added to our discussion as shown in the attachment.

2. Thank you for pointing this out. We did not intend to create confusion by writing about MRI in this section. We discussed MRI along with selected additional imaging modalities in the spirit of offering alternatives to interested readers. We have used MRI in conjunction with our experimental model in the past, but currently prefer pressure-volume loop catheterization because this technique is uniquely able to investigate load-independent properties of the heart, such as myocardial contractility. We have clarified this, as follows: Although echocardiography and cardiac MRI clearly possess strong advantages and have been employed by our lab in the past (see Table 1), we no longer use these techniques, and have arrived instead in recent years at pressure-volume loop catheter-based trandusction of myocardial functional parameters as our preferred analytic method. This is because, while the desirability of longitudinal functional analysis should not be understated, the concurrent acquisition of pressure and volume-based parameters permits a load-independent assessment of cardiac chamber mechanics that cannot be obtained by other means.82 We further appreciate your point regarding the desirability of focusing on ameroid constriction. In addition to the discussion of the physiology of ameroid constriction covered in the section on establishing chronic myocardial ischemia, we have added further information to the Procedure section about placement principles and troubleshooting, along with an additional video directly depicting ameroid constrictior placement. 

3. We have added an additional video depicting ameroid constriction (Video 9), which we will request that the editorial staff make available. We have also added substantial additional description to the ameroid placement procedure, as delineated in the attached file.

4. Thank you for these suggestions. We have taken them up, describing neutron activation in further detail as follows: In all cases, isotopic microsphere analysis is achieved using neutron activation, in which isotopes are rendered active following tissue collection and thus emit radiation that can be precisely quanititated. Neutron activation entails exposure to a neutron field, precipitating brief gamma emission by microsphere atoms and subsequent spectroscopic analysis with quantification of disintegrations per minute. It does not require destruction of the tissue, raising the possibility of additional subsequent analysis if needed. This technique is highly sensitive, with the capacity to detect even a single microsphere embedded in analyzed tissue. Additionally, it confers the substantial advantage over the previous gold standard of radioactive microspheres of mitigating exposure of lab personnel to the consequent occupational hazards, and over the optical technique previously used by our laboratory of obviating additional tissue processing.72,73,74 Additional methodological description of our injection procedures were also included in the Procedure section, and appended to the attached file.

5. Thank you for highlighting the need for clarification of this aspect of our protocol. Using our microsphere perfusion studies, we are able to precisely quantify the region of the heart affected by ischemia. Briefly, we divide the heart after resection sixteen times. Because we injected gold mircrospheres into the left atrial appendage during our first surgical procedure while occluding the LCx coronary artery, the most ischemic region of the heart will be the section of the heart with the lowest count of gold microspheres. We then use immunohistochemistry to stain this section of the heart for arteriolar and capillary growth, thus providing precise quantification of ischemia and angiogenesis in this segment. Analogously, the other microspheres we use (samarium and lutetium) allow us to quantify perfusion of all sections of the heart according to location, which can then be used to compare treated hearts with control hearts. In this way, we are able both to confirm ischemia and to test relative perfusion highly effectively with our model. Added information to this effect has been placed in our manuscript and represented in the attached file.

6. Thank you for asking this question, as it provides an opportunity to enrich the discussion within our text of relevant porcine anatomy. Access to sternal saws is not an issue, but the porcine thorax, unlike the heart it contains, possesses significant anatomic deviation from that of humans. In particular, it possesses a 90-degree deviation and is extremely thick at its most superior aspect, precluding the use of a sternal saw as in humans. Other investigators have approached this issue by opening the chest off midline, but we prefer the Lebschke knife for this purpose, as it allows easy adjustment and verification of freedom from impingement on the great vessels. We have included information to this effect in the revised text. See the following reference: doi: 10.1016/j.jaccas.2022.06.011.

Round 2

Reviewer 3 Report

Comments and Suggestions for Authors

I want to thank the authors for their response to my comments. The manuscript improved signifcantly.

Comments on the Quality of English Language

Minor editing of the English language required.